# Diesel2p mesoscope with dual independent scan engines for flexible capture of dynamics in distributed neural circuitry

Che-Hang Yu[1,3], Jeffrey N. Stirman[2,3], Yiyi Yu [1], Riichiro Hira [1] & Spencer L. Smith [1✉]

Imaging the activity of neurons that are widely distributed across brain regions deep in scattering tissue at high speed remains challenging. Here, we introduce an open-source system with Dual Independent Enhanced Scan Engines for Large field-of-view Two-Photon imaging (Diesel2p). Combining optical design, adaptive optics, and temporal multiplexing, the system offers subcellular resolution over a large field-of-view of ~25 mm$^2$, encompassing distances up to 7 mm, with independent scan engines. We demonstrate the flexibility and various use cases of this system for calcium imaging of neurons in the living brain.

[1] Department of Electrical and Computer Engineering, University of California, Santa Barbara, CA, USA. [2] LifeCanvas Technologies, Cambridge, MA, USA. [3] These authors contributed equally: Che-Hang Yu, Jeffrey N. Stirman. ✉email: sls@ucsb.edu

Two-photon microscopy[1] has enabled subcellular resolution functional imaging of neural activity deep in scattering tissue, including mammalian brains[2]. However, conventional microscopes provide subcellular resolution over only small fields-of-view (FOVs), ~Ø0.5 mm. This limitation precludes measurements of neural activity distributed across cortical areas that are millimeters apart (Fig. 1a). Obtaining subcellular resolution over a large FOV involves scaling up the dimensions of the objective lens and other optics, due to the Smith-Helmholtz invariant, also known as the optical invariant[3–6]. However, that is only half of the solution. Since high light intensities are required for efficient multiphoton excitation, two-photon imaging is typically implemented as a point-scanning approach, where an excitation laser beam is scanned over the tissue. Thus, each voxel sampled entails a time cost, and the scan engine design constrains the temporal resolution[7]. Temporal multiplexing of simultaneously scanned beams can increase throughput[8], and these can have either a fixed configuration[9–11], or can be reconfigured during the experiment axially[12–14] or axially and laterally[15,16]. However, these simultaneously scanned, or "yoked", multi-beam configurations strongly constrain sampling, because they preclude varying the scan parameters among the multiplexed beams. Optimal scan parameters (e.g., frame rate, scan region size) vary across distributed neural circuitry and experimental requirements, but yoked scanning requires using the same scan parameters for all beams. Therefore, a system featuring both a large imaging volume and independent multi-region imaging is needed, and can enable new experiments.

In this work, we develop a system with a large FOV, subcellular resolution, and dual independent scan engines for highly flexible, asymmetric multi-point sampling of distributed neural circuitry. Here, we present a custom two-photon system with dual scan engines that can operate completely independently. Each arm has optical access to the same large imaging volume (~25 mm² FOV) over which subcellular resolution is maintained in scattering tissue to typical 2-photon imaging depths. These two arms use adaptive optics (AO) for wavefront shaping, temporal multiplexing for simultaneous imaging, and polarization optics for beam recombination. Due to the independence of the arms, and the use of polarization optics for beam combining, the input lasers can come from the same source or different sources (multiwavelength). Moreover, each arm can use multiple sources simultaneously, for example, in imaging and photoactivation experiments. We refer to the system as the Diesel2p (Dual Independent Enhanced Scan Engines, Large field-of-view Two-Photon).

## Results

**System design and performance benchmarks**. The Diesel2p has two major design features. First, it has a ~25 mm² FOV to encompass multiple cortical areas and provides subcellular resolution throughout (Fig. 1a). Second, the Diesel2p can perform simultaneous two-region imaging using two scan engine arms. In contrast to prior work[12,15], these two arms are completely independent. They can each scan any region and be configured to with different imaging parameters (e.g., pixel dwell time, scan size) including random access scanning (Fig. 1b–d). To achieve these two features, several scan engine components were custom designed and manufactured: the optical relays, the scan lens, the tube lens, and the objective. The full optical prescriptions are provided in this report (Supplementary Fig. 1–5, ZEMAX models). The system was optimized as a whole, rather than optimizing components individually, to minimize the aberrations across scan angles up to ±5 degrees at the objective back aperture and primarily for the excitation windows of 910 ± 10 nm and

1050 nm ± 10 nm. The optics use an infinity-corrected objective design to facilitate modifications and modularity. Based on the optical design model, the Strehl ratio exceeded 0.8 (consistent with a diffraction-limited design) over an area only slightly smaller than a 5 × 5 mm² (25 mm²) square, which is ~28% larger than a 5 mm diameter circle (Fig. 1e).

The Diesel2p system uses two independent scan engines to access two areas simultaneously (Fig. 1a, b, d), as opposed to one beam jumping back and forth between two areas sequentially. In the sequential imaging regime, information is missed both during the scanning of the other area and during the time of jumping. This latter dead time is a larger fraction of the duty cycle as the frame rate increases (Supplementary Fig. 6). The laser beam, after passing through the dispersion compensator[17], is split into two pathways by a polarization beam splitter (Fig. 1b). The temporal multiplexing is set by delaying one laser beam's pulses relative to the other (for an 80 MHz system as used here, the delay is 6.25 ns). Beams are guided into two independent scan engines. Each scan engine consists of an x-resonant mirror, an x-galvo mirror, and a y-galvo mirror in series, each at conjugate planes connected by custom afocal relays. The x-resonant mirror provides rapid and length-variable x-line scanning, up to 1.5 mm. The x- and y-galvo mirrors provide linear transverse scanning across the full FOV. Therefore, each arm of the scan engine can arbitrarily position the imaging location within the full FOV, and scan with parameters that are completely independently from the other arm. Each pathway is also equipped with a deformable mirror AO for both rapidly adjusting the focal plane axially (Supplementary Fig. 7a and Supplementary Video 1) and correcting optical aberrations (Supplementary Fig. 7b, c and Supplementary Video 2).

Next, we measured the resolution of the Diesel2p system by taking z-stacks of 0.2-μm fluorescent beads at various positions and depths. The lateral and axial resolutions were estimated from the full-width-at-half-maximum (FWHM) of measured intensity profiles. For beads at each XYZ location in the FOV, measurements were made with the deformable mirror optimized to act as an AO element. Overall, the lateral FWHM was ~1 μm and axial FWHM was ~8 μm across the 5-mm FOV and up to 500-μm imaging depth (Fig. 1f). This indicates a space-bandwidth product of ~(25 mm²/1 μm²) = 25 × 10⁶, or 25 megapixels. The use of the deformable mirror as an AO element reduced the resolution variation across the measured volume, and improved the axial resolution by ~2 μm (Supplementary Fig. 8). The AO also enabled imaging of neural activity over 3.5 mm from the center of the FOV, equivalent to a 7-mm diameter along the diagonal axis (Supplementary Fig. 7b). These results show that the Diesel2p system maintains a nearly constant subcellular resolution throughout the 25 mm² FOV, and allows measurement of neural activity across this area.

For efficient multiphoton excitation, the characteristics of the ultrafast pulses must be maintained over the full FOV. We measured pulse characteristics at the focal plane using the frequency-resolved optical gating (FROG) technique[18]. Throughout the FOV, the pulse width is maintained at ~110 fs, and the pulse front tilt and the spatial chirp remain low (Supplementary Fig. 9). Together with the bead measurement, these results show that the Diesel2p system has a nearly consistent resolution and spatiotemporal pulse characteristics across the entire FOV.

We next verified the imaging FOV by imaging a structured fluorescent sample with repetitive 5 lines per mm (57–905, Edmund Optics). Images contain 25 lines along both the x and y directions, indicating a 5-mm length on each axis of the FOV (Fig. 1g). The result demonstrates that the Diesel2p system has a FOV very close to a 5 × 5 mm² field, consistent with the nominal model performance (Fig. 1e). By z-scanning through this sample

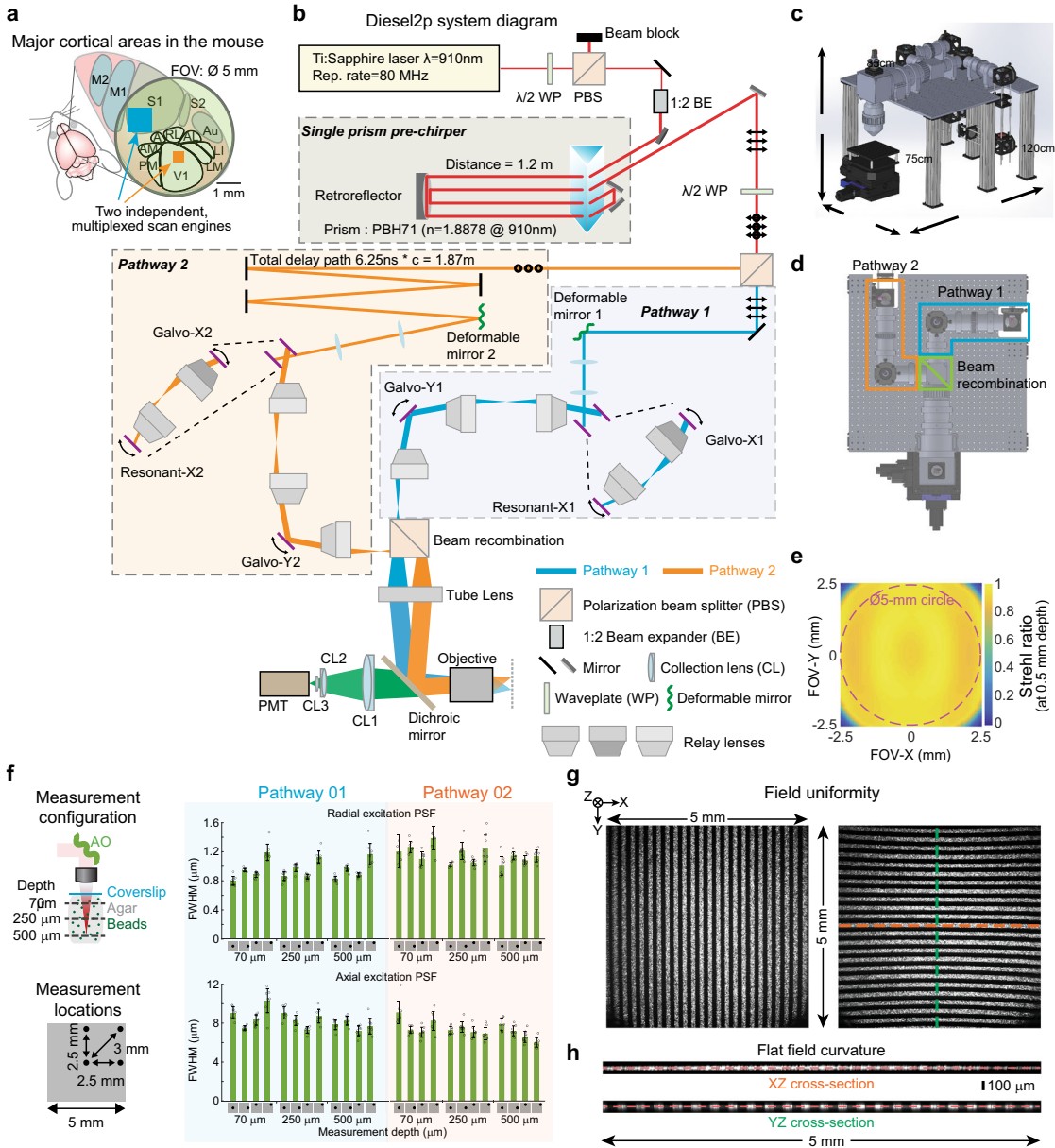

**Fig. 1 Diesel2p system features, layout, and performance benchmarks. a** Functional cortical areas in the mouse brain are widely distributed. A field-of-view (FOV) of Ø5 mm can encompass multiple brain areas, and independent scan engines can capture ongoing neural activity in multiple cortical areas simultaneously with optimized scan parameters. **b** Two imaging beams are temporally multiplexed and independently positioned in XY using two sets of resonant-galvo-galvo scan engines. First, overall power is attenuated using a half-wave plate (λ/2 WP) and a polarizing beam splitting cube (PBS). A 2X beam expander (1:2 BE) enlarges the beam for the clear aperture of the deformable mirrors adaptive optics (AO). A custom single-prism pre-chirper offsets system dispersion to maintain transform-limited pulses at the focal plane. A second λ/2 WP and PBS pair divides the beam into two pathways. Pathway 2 (s-polarization in orange) passes to a delay arm where it travels 1.87 m further than Pathway 1 using mirrors, thus delaying it by 6.25 ns relative to Pathway 1 (p-polarization in blue). Both pathways each proceed to deformable mirrors for adjusting the focal plane and correcting optical aberrations before being directed to resonant-galvo-galvo scan engines. All scanning mirrors are optically relayed to each other. Each pathway then passes through a scan lens before being combined with a beam recombination relay. A tube lens and an infrared-reflective dichroic mirror relay the two multiplexed beams onto the back aperture of the objective. Fluorescence (green) is directed to a photomultiplier tube (PMT) via an assembly of collection lenses (CL1, CL2, CL3). **c** An oblique view of a 3-D model of the system and its footprint. **d** A top view with the arrangement of the two scan engines highlighted. **e** Plot of the model Strehl ratio across the scan area indicates diffraction limited performance (Strehl ratio > 0.8) across ~25 mm$^2$, significantly larger than the area of the dashed 5-mm diameter circle (~19.6 mm$^2$) by ~28%. **f** Multiphoton excitation PSF measurements were made with subresolution beads (0.2 μm) in agar under a coverslip at three depths and four locations, for both of the AO-equipped, temporally multiplexed beam pathways. FWHM of the Gaussian fits for measurements from the fluorescence beads radially and axially are calculated and plotted. Eight beads ($n = 8$) at each locations are measured, except that there are 7 beads ($n = 7$) measured on axis at the depth of 500 μm. Data are presented as mean values ± S.D. **g** XY images of a calibrated, structured fluorescent sample with a periodic line pattern (5 lines per millimeter) in two orientations acquired under the full scan range of the system. Each image shows 25 lines on the top edge (left image) and on the left edge (right image), receptively, verifying a 5 × 5 mm FOV. **h** The XZ image along the orange dashed line and the YZ image along green dashed line in (**g**) are also plotted. The imaging pattern is colinear with the straight lines, suggesting a flat field both in x and y directions across the FOV.

and rendering the x-z and y-z cross-section, we found that the thin fluorescence pattern is nearly co-linear with a straight line, indicative of a field curvature <30 μm over 5000 μm of FOV (Fig. 1h). This result also demonstrates that the Diesel2p system has the flattest field among the reported mesoscopes with a FOV of 5 mm diameter or beyond[3,19]. For the application of in vivo neuronal imaging, this extent of field curvature is negligible, and field curvature calibration and correction are not necessary. Together, these benchmarks show that the system exhibits subcellular resolution throughout a flat 25 mm$^2$ FOV for both imaging pathways.

**Two-photon imaging in vivo**. After benchmarking the optical performance imaging system, we performed a series of in vivo imaging experiments with neurons expressing the genetically encoded calcium indicator, GCaMP6s[20]. The brain tissue under a 5 mm diameter cranial window was positioned within the $5 \times 5$ mm$^2$ FOV, and subcellular detail was resolved in individual neurons across the FOV (Fig. 2a). To further verify the Diesel2p's subcellular resolving power, we also performed in vivo imaging of dendritic spines (in a Thy1-GFP mouse), which were also resolved >250 μm deep (Fig. 2b). This result demonstrates subcellular resolution in scattering, living brain. Next, we positioned the pathways to image two adjacent stripes of cortex simultaneously and set each pathway to scan a large FOV of $1.5 \times 5$ mm$^2$ ($1024 \times 4096$ pixels), of an awake mouse (Fig. 2c and Supplementary Video 3). In this data set, we imaged a total area of 15 mm$^2$ with a pixel resolution of ~$1.5 \times 1.2$ μm$^2$ (undersampling the resolution for the sake of increased frame rate), and an imaging rate of 3.84 frames/s, resulting in a pixel throughput of 32.3 megapixels/s over 15 mm$^2$. Calcium signals from 5,874 neurons were detected from these two stripes (Fig. 2d). The raw calcium signals had a signal-to-noise ratio of $7.9 \pm 2.5$ (mean ± standard deviation) (Fig. 2e), and this supported robust spike inference (Fig. 2f), which we used to calculate the correlation matrix and plot how correlations vary as a function of distance between neuron pairs (Fig. 2g, h). The correlations are relatively high at distances less than 40 μm, and there is a slow decrease beyond 40 μm, and the measurements supported this plot out to 4000 μm. This data set demonstrates the ability of the system to measure neuronal activity with high fidelity over the large FOV.

**Flexible measurement with dual independent scan engines**. To demonstrate the flexibility of the Diesel2p system, we performed four test experiments. First, we imaged two regions that were 4.36 mm apart, which is equivalent to the distance between the primary visual cortex and the motor cortex. Moreover, we configured the imaging fields, frame acquisition speeds, and the pixel numbers independently for the two regions (Fig. 3a and Supplementary Video 4). Non-multiplexed imaging with these parameters would reduce the acquisition rate and involve >20% dead time (Supplementary Fig. 6), and yoked multiplexing would require a compromise in imaging parameters so that both regions would have had the same size and scan rate. Thus, the Diesel2p system enables new classes of measurements. Second, to demonstrate that the two pathways can be overlapped, we set Pathway 1 to image a subregion of the region imaged by Pathway 2 (Fig. 3b and Supplementary Video 5). Third, we increased the number of imaging regions within a single imaging session to four. Two sub-regions are imaged simultaneously, then they are both repositioned (this entails time for repositioning the beams, or a "jump" time). In this way, a total of four sub-areas that differed in XY locations and Z depths were imaged in a single imaging session (Fig. 3c and Supplementary Video 6). Fourth, we

used random-access scanning of cell bodies in conjunction with large FOV imaging. We configured Pathway 1 to raster scan a 2.25 mm$^2$ area, and Pathway 2 executed a random-access scan of 12 cell bodies sequentially (Fig. 3d and Supplementary Video 7). Finally, we characterized the crosstalk between the two imaging pathways, and found it to be minimal (Supplementary Fig. 10). Together, these results, enabled by both the large FOV and the dual independent multiplexed scan pathways, demonstrate the flexibility of the Diesel2p system to enable new measurements and experiments.

## Discussion

In summary, we present a newly developed imaging system to enable flexible, simultaneous, multi-region, multiphoton excitation in scattering tissue. The Diesel2p system has a nearly constant subcellular resolution over a ~25 mm$^2$ FOV with very low field curvature, linear galvo access to the full FOV, and resonant scan size of 1.5 mm. These optics and scan specifications are combined with a layout of two independent scan engines, each with deformable mirrors for fast z-focus and aberration correction. The temporally multiplexed imaging pathways can record neural activity in two arbitrarily selected portions of the imaging volume simultaneously.

This new system is also designed to facilitate experiments with behaving animals. The objective lens is air immersion, so no water interface is required, and it has an 8-mm-long working distance, to enable a variety of headplate designs and other instrumentation that needs to be close to the imaged area (e.g., electrode arrays). In addition, the objective can rotate 360 degrees to work with non-horizontal imaging surfaces. This rotation facilitates imaging in some preparations and could be further improved if the axis of rotation was along the focal plane of the objective. The advantage of an air immersion objective is evident when the objective is fixed at an angle for imaging the brain of a behaving animal from the side. These ergonomic features can facilitate behavior experiments that require flexibility for animal posture, further increasing the flexibility of the Diesel2p system.

The Diesel2p system uses an infinity-corrected objective design, making it compatible with a range of extensions to multiphoton imaging, including Bessel-beam scanning[21] and reverberation microscopy[14] to enhance the volumetric imaging capability. It can also be combined with two-photon optogenetics approaches to perform simultaneous multiphoton neuronal imaging and functional perturbation[22]. Wavelength multiplexing can be implemented to add another pulsed laser (e.g. a 1040 nm pulsed laser) into the Diesel2p system, enabling the dual-excitation imaging of two different molecules simultaneously (Supplementary Fig. 11). The Diesel2p system is achromatic at 910 nm and 1040 nm, and thus can work simultaneously with these two wavelengths with no need of realignment such as spacing between optics. The advantage of the Diesel2p's simultaneous imaging, with zero jumping time, can be critical when imaging very fast dynamics such as neurotransmitter reporters[23] and voltage indicators[24] expressed at distant brain areas. The Diesel2p system enables new measurements of neural activity, is compatible with a range of variants of multiphoton imaging, and is a fully documented and open optical design that can be extended to support studies of neural interactions across brain areas[25].

## Methods
**Optical design and simulations**. The entire system including relay, scan, tube, and objective lens systems were modeled in OpticStudio (Zemax, LLC). The subsystems of relay, scan, and tube, and objective lenses were first designed, modeled, and optimized individually. Then, the system was optimized as a

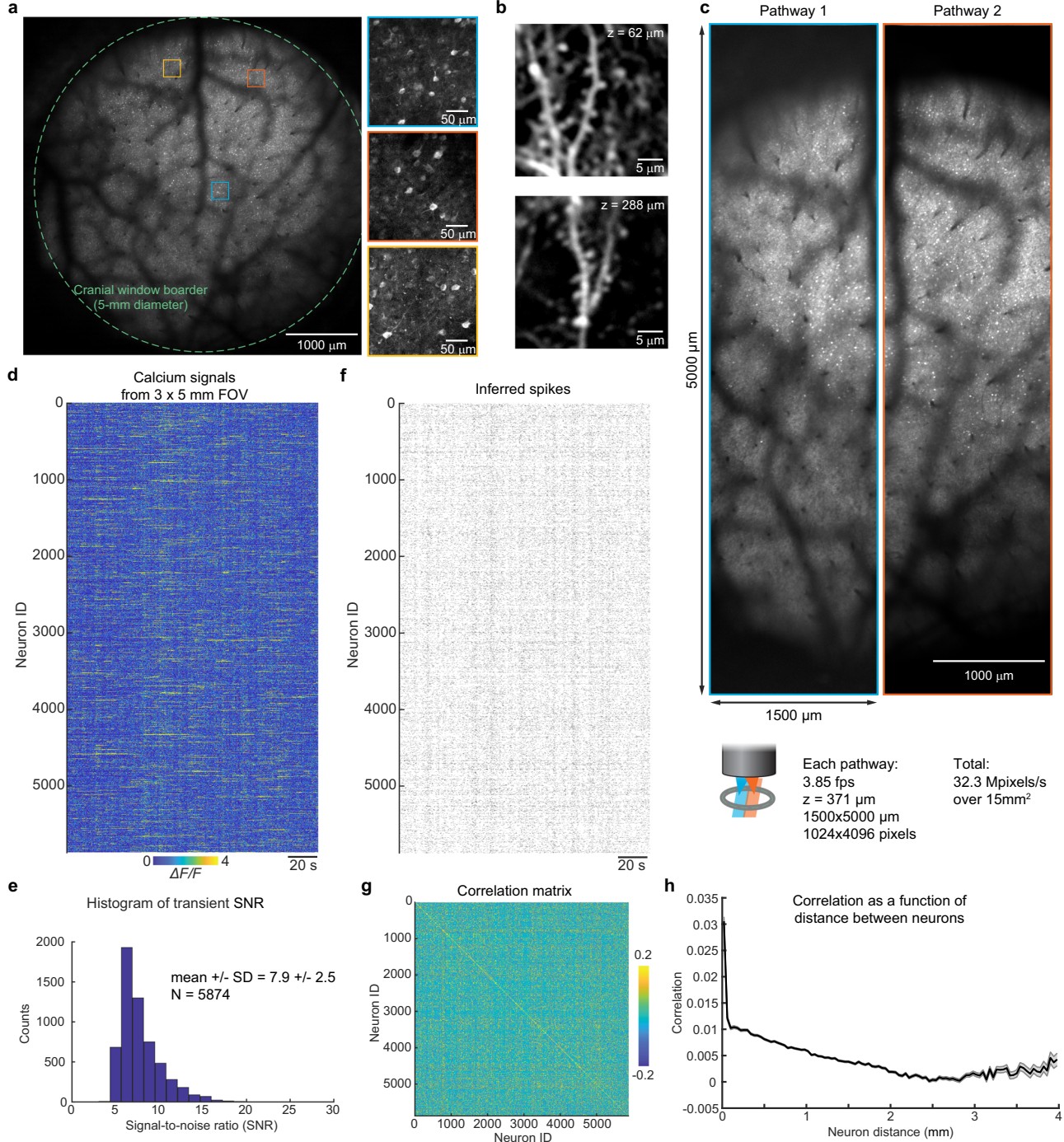

**Fig. 2 Diesel2p provides subcellular resolution two-photon imaging of neural activity across its FOV. a** Diesel2p's galvo-galvo raster scanning records neuronal activity at the depth of 345 μm through a cranial window with a diameter of 5 mm (dashed line) on a transgenic mouse expressing the genetically encoded fluorescent calcium indicator GCaMP6s in excitatory neurons. Three zoom-in views from different sub-regions (colored squares) show the preservation of the subcellular resolution. **b** Dendritic spines were resolved in vivo from a transgenic mouse expressing Thy1-GFP at different depths. **c** Neuronal activity in two strips of cortex are recorded using Diesel2p's two pathways simultaneously (blue and orange rectangles), covering a combined area of 3 mm × 5 mm with a frame rate of 3.85 frames/s. The imaging depth is 371 μm. The pixel number for each image is 1024 × 4096 pixels. **d** In this data set, calcium signals were recorded from 5,874 active neurons. **e** The histogram of the transients' signal-to-noise ratio in **d**. **f** $Ca^{2+}$ signals in **d** were used to infer spike times. **g** Spikes in **f** were used to measure >17 million cross-correlations per experiment. **h** The correlations in **g** plotted as a function of distance between neurons ranging from 0 μm to 4000 μm. Data are presented as mean values ± S.D.

whole to further minimize additive aberrations between subsystems. The system was optimized for two wavelengths of 910 ± 10 nm and 1050 ± 10 nm for dual-color imaging in the future. All lenses were custom made by Rocky Mountain Instrument Inc, using tolerances of: radii ±0.1%, center thickness ±0.1 mm and coated with broad band anti-reflective coating (BBAR), $R_{avg} < 1.5\%$ at 475nm-1100 nm. The relay lenses between the x-resonant scanner and the x-galvo mirror were constructed from the Thorlabs Inc parts (LSM254-1050.ZBB, AC508-250-B-ML). The effective focal lengths of the custom scan lens, the tube lens, and objective are 61 mm, 243 mm, 30 mm, respectively. Complete lens data of the Diesel2p system is given in Supplementary Figs. 1–5.

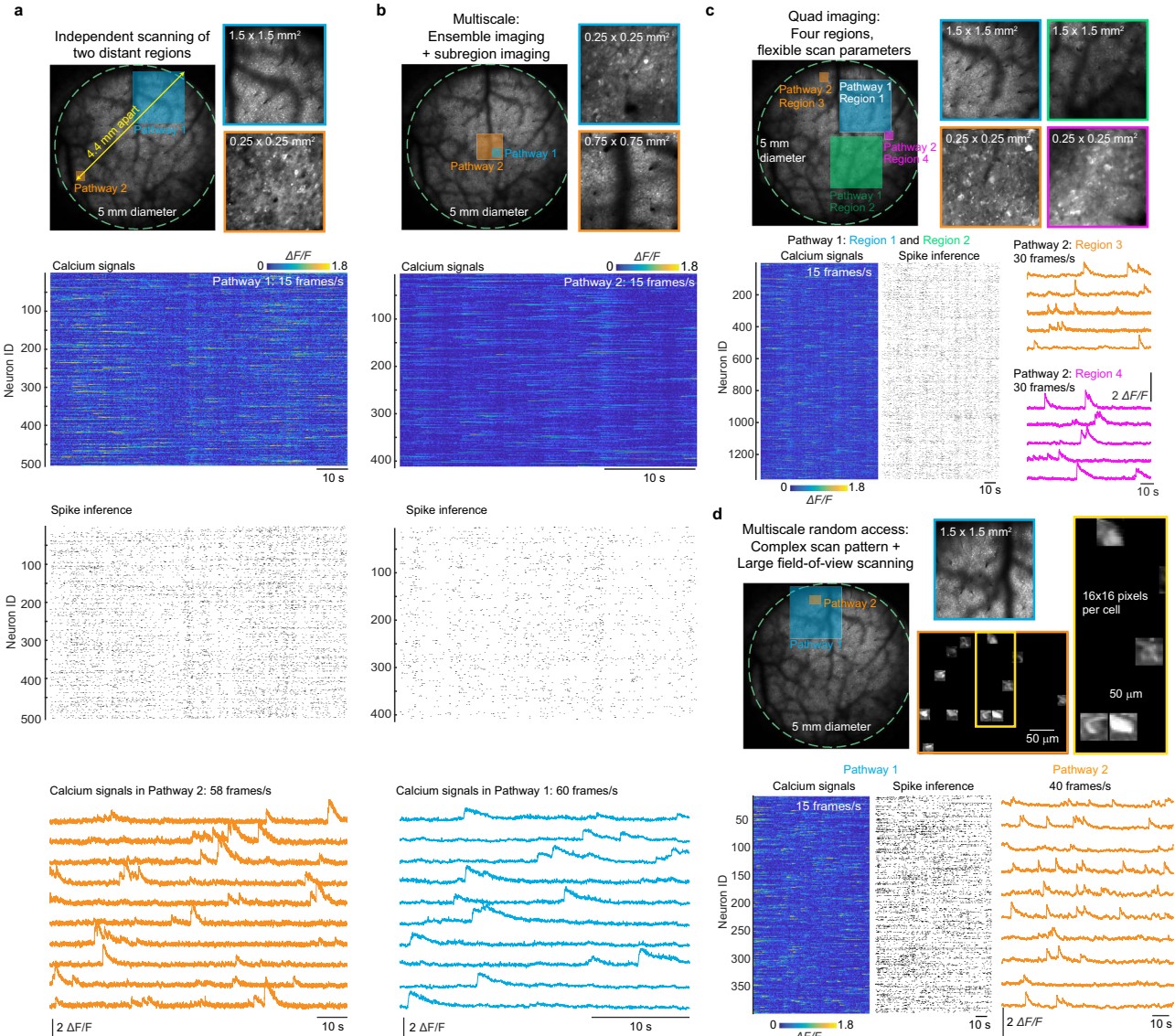

**Fig. 3 Flexible measurement with dual independent scan engines. a** Neuronal activity at two distant regions (4.36-mm apart) are imaged simultaneously with independent imaging parameters. The blue and orange boxes indicate the imaging sizes and positions within the full FOV. Expanded regions are shown at right. Calcium signals from 500 neurons imaged in Pathway 1 were used to infer spikes. Simultaneously, Pathway 2 imaged calcium signals at 58 frames/s 4.36 mm away. **b** Neuronal activity from two overlapped regions are imaged simultaneously with different imaging parameters. Over 400 neurons were imaged in Pathway 2 while neural activity was imaged at 60 frames/s in Pathway 1. **c** Neuronal activity from four separate regions are imaged with two pathways independently positioned and then repositioned within the full FOV. By serially offsetting the galvo scanners, Pathway 1 accesses the blue and green regions, and Pathway 2 accesses the orange and the magenta regions. By changing the curvature of the AO, Pathways 1 and 2 also image at different depths. Calcium signals and inferred spike trains from neurons in the Pathway 1 data (Region 1 and Region 2 combined) are shown. Example calcium signals are shown for neurons from Region 3 (orange) and Region 4 (magenta) scanned by Pathway 2. **d** A combination of raster scanning and random-access scanning is configured on Pathways 1 and 2 for neuronal imaging. While Pathway 1 performs raster scanning, Pathway 2 performs random-access scanning for 12 cell bodies. Calcium signals and the inferred spike trains are shown for neurons imaged by Pathway 1. Example calcium signal traces are shown for neurons imaged by Pathway 2 (orange).

**Assembly**. The lens sub-assemblies were manufactured, aligned, and assembled in the factory of Rocky Mountain Instrument Co. There are threaded connectors between optical relays connected the two galvo scanners, and between the tube lens and the objective lens. Together with the correction collar on the objective, they are adjustable for the axial separation between subassemblies. Galvo and resonant scanners were mounted on a XY translator (Thorlabs, CXY2), and this was attached to a 60 mm cage cube (Thorlabs, LC6W), which also bridged the sub-assemblies. As the system is assembled, locations of afocal space at conjugate planes are checked for collimation as designed in the system.

**Animals**. All procedures involving living animals were carried out in accordance with the guidelines and regulations of the US Department of Health and Human Services and approved by the Institutional Animal Care and Use Committee at University of California, Santa Barbara. We used GCaMP6s and Thy1-GFP Line O (Jackson Labs stock #007919) transgenic mice in this study. GCaMP6s transgenic mice were generated by triple crossing of TITL-GCaMP6s mice, Emx1-Cre mice (Jackson Labs stock #005628) and ROSA:LNL:tTA mice (Jackson Labs stock #011008)[26]. TITL-GCaMP6s mice were kindly provided by Allen institute. Mice were housed in 12 h dark/light reverse cycle room. The temperature set-point is 74–76 F; the low-temperature alarm is 70 F; the high-temperature alarm is 78 F. The relative humidity is ~45% (range 30–70%). Mice were deeply anesthetized using isoflurane (1.5–2%) augmented with acepro-mazine (2 mg/kg body weight) during craniotomy surgery. Carpofen (5 mg/kg body weight) was administered prior to surgery, as well as after surgery for 3 consecutive days. A 5 mm diameter cranial window was implanted after removing the scalp overlaying the right visual cortex.

**In vivo two photon imaging**. All imaging was performed on the custom Diesel2p system. The instrumentation (see Diesel2p instrumentation below) and image acquisition were controlled by ScanImage from Vidrio Technologies Inc. Animals were awake during calcium imaging. The imaging was performed with <100 mW out of the front of the objective. With typical imaging parameters (512 × 512 at 30 frames/s, 0.5 mm imaging region), no damage was observed from the surface of the dura to the 500 μm depth. Assessment of damage due to laser intensity was based on visual morphological changes to the appearance of the dura mater and/or continuously bright cell bodies.

**Pulses per pixel in the resonant scanning regime**. When raster-scanning with a resonant mirror, the pulses per pixel along a line scan varies due to the nonlinear scanning speed of the resonant scanner. The pulses per pixel is a function of the fill fraction (FF), the number of pixels per line (N), the resonant frequency of the resonant scanner (Freq), and the repetition rate of the laser (Rep), and the position of the pixel on the resonant axis (n). The ratio between the length of an active acquisition of a line and the total length of the line is defined as the fill fraction. Equation (1) shows the formula to calculate the pulses per pixel as follows.

$$Pulses\ per\ pixel(n, FF, N, Freq, Rep) = \frac{2*FF}{N} * \frac{Rep}{2*\pi*Freq*\cos\left\{\sin^{-1}\left[-FF + \frac{2*FF}{(N-1)}(n-1)\right]\right\}} \tag{1}$$

Three commonly used imaging parameters are plotted in Supplementary Fig. 12.

**Image analysis for neuronal calcium signals**. $Ca^{2+}$ signals were analyzed using custom software[27] in MATLAB (Mathworks). Neurons were segmented and fluorescence time courses were extracted from imaging stacks using Suite2p (https://suite2p.readthedocs.io/en/latest/)[28]. Signals from neurons are a sum of neuronal and neuropil components. The neuropil component was subtracted from the neuronal signals by separately detecting it and subtracting it. The neuropil component was isolated using the signal from an annulus region around each neuron, and then subtracted from the neuronal signal to provide a higher fidelity report of neuronal fluorescence dynamics. Subsequently, spike inference was performed on these neuropil-subtracted traces using a Markov chain Monte Carlo method[29]. The parameters for the MCMC spike inference were p = 2 (second order autoregressive model), b = 200 (initial burn), 400 rounds of simulation, and then the frame rate for each data set. We computed the Pearson correlation of inferred spike trains between neurons using MATLAB built-in function "corr".

**Excitation point spread function measurements and simulations**. The measurement and analysis procedure were described in our previous publication in details[15]. To evaluate the excitation point spread function (PSF), sub-micrometer beads were imaged. Sub-micrometer fluorescent beads (0.2 μm, Invitrogen F-8811) were imbedded in a thick (~1.2 mm) 0.75% agarose gel. 30 μm z-stacks were acquired, each centered at one of three depths (50 μm, 250 μm, 500 μm). The stage was moved axially in 0.5 μm increments ($\Delta_{stage}$). At each focal plane 30 frames were acquired and averaged to yield a high signal-to-noise image. Due to the difference between the refractive index of the objective immersion medium (air) and the specimen medium (water), the actual focal position within the specimen was moved an amount $\Delta_{focus} = 1.38 \times \Delta_{stage}$[30]. The factor 1.38 was determined in Zemax and slightly differs from the paraxial approximation of 1.33. These z-stack images were imported into MATLAB for analysis. For the axial PSF, XZ and YZ images were created at the center of a bead, and a line plot was made at an angle maximizing the axial intensity spread, thereby preventing underestimation of the PSF due to tilted focal shifts. For the radial PSF, an XY image was found at the maximum intensity position axially. A line scan in X and Y was made. Gaussian curves were fit to the individual line scans to extract FWHM measurements. The radial PSF values are an average of the X PSF and Y PSF, and the axial PSF is an average of the axial PSF found from the XZ and YZ images. Excitation PSF measurements were performed both on axis and at the edges of the FOV for both imaging pathways. Data reported (Fig. 1i and Supplementary Fig. 8b) are the mean ± S.D. of eight beads.

**Diesel2p instrumentation**. Our laser source is a Ti:sapphire pulsed laser with a central wavelength at 910 nm and a 80 MHz repetition rate (Mai-Tai, Newport). The laser first passes through a built-in pre-chirper unit (DeepSee, Newport), further followed by an external custom-built single prism pre-chirper[17]. The material of the prism is PBH71 (Swamp Optics) with a refractive index of 1.89. The laser power is controlled using a half wave plate (AHWP05M-980, Thorlabs) followed by a polarization beam splitting cube (CCM5-PBS203, Thorlabs). Similar polarization optics were used to split the beam into two paths and control the relative power between the two paths. Prior to splitting, the beam was expanded using a 2× beam expander (GBE02-B, Thorlabs). One beam travels directly to a deformable mirror (DM140A-35-UM01, Boston Micromachines Corporation), and the other beam is first diverted to a delay arm, and subsequently to a separately deformable mirror. The delay arm is designed to impart a 6.25 ns temporal offset to the pulses in one beam (1.875 m additional path length). As the laser pulses are

delivered at 12.5 ns intervals (80 MHz), they are evenly spaced in time at 160 MHz after the two beams are recombined. Before the recombination, both pathways pass through their own set of scan engines comprised with a x-resonant scanner (CRS8KHz, Cambridge technology), and a x-galvo scanner (6220H, Cambridge technology), and a y-galvo scanner (6220H, Cambridge technology) in series. These scanners are connected by custom-designed afocal relays. Two beams are recombined with another polarization beam splitter (PC75K095, Rocky Mountain Instrument). A scan lens and tube lens formed a 4× telescope. Together with a short-pass dichroic mirror, they relayed the expanded beams to the back aperture of the custom objective. The entrance scan angles at the objective back aperture were ~±5 degrees yielding our 5 mm FOV. The generated fluorescence from the imaging plane is directed to a photomultiplier tube (PMT, H10770PA-40 MOD, Hamamatsu) via the assembly of 3 collection lenses (AC508-100-A-ML, AC254-030-A-ML, Thorlabs; 48425, Edmunds Optics). The ultrafast 1040 nm laser used for dual-wavelength imaging was an ALTAIR IR-10 (Spark Lasers). The optics are achromatic for the 910 nm and 1050 nm wavelength windows, so both wavelengths can be used simultaneously in either or both paths without any reconfiguration of the imaging system.

**AO optimization**. We adopted a sensorless and model-based approach for the wavefront correction. Eleven of the first 15 modes of Zernike coefficients are corrected sequentially and manually to maximize the brightness of the image. The modes of piston, tip, tilt, and defocus are not adjusted. In general, three iterations of adjustment across the 11 modes reaches a plateau of brightness. Compensation is location dependent. An optimal configuration at one area is not applicable to other areas. Look-up-table approaches could be used to vary corrections during scanning due to the fast update of deformable mirrors (~0.8–1.5 ms).

When the AO serves as a remote focusing component, only the defocusing term of the Zernike coefficients is adjusted to move the imaging plane. The current maximum defocusing range is 120 μm (± 60 μm from the middle plan) limited by the 3.5 μm maximum stroke traveling distance of the Boston Micromachines deformable mirror. The ALPAO unit has a stroke of 80 μm, which is almost 23-fold more, and thus can offer a greater defocus range.

**Photon counting electronics**. Output from the photomultiplier tube was first amplified with a high bandwidth amplifier (C5594-44, Hamamatsu) and then split into two channels (ZFSC-2-2A, Mini-Circuits). One channel was delayed relative to the other by 6.25 ns by using a delay box (DB64, Stanford Research Systems). Each channel was connected to a fast discriminator (TD2000, Fast ComTec GmbH). The ~80 MHz synchronization output pulses from the laser was delivered to a third fast discriminator (TD2000, Fast ComTec GmbH), which has a continuous potentiometer adjustment to adjust the output pulse width from 1 to 30 ns. This output pulse was delivered to the common veto input on the previous two TD2000 discriminators where the PMT outputs were collected. The veto width was adjusted by the potentiometer on third TD2000 discriminator and the relative phase of the veto window was adjusted by delaying the synchronization pulses from the laser module using the DB64 delay box. Outputs from each TD2000 is sent to a channel of the digitizer of the vDAQ card (Vidrio Technologies). Digitized signals were arranged into images with the indicated pixel count in the ScanImage software (Vidrio Technologies). In this manner we could demultiplex the single PMT output into two channels corresponding to the two excitation pathways.

**Pulse characterization at the focal plane**. We used the frequency-resolved optical gating (FROG) method measurements to retrieve the pulse conditions at the focal plane using three off-axis parabolic (OAP) mirrors and a FROG system (GRENOUILLE 8-50-334-USB, Swamp Optics). We used reflective OAP mirrors to avoid the post-focus chromatic and spatial dispersion that would be introduced if refractive lenses were used. OAP mirrors help retain more of the original pulse information of the pulses. The focused laser at the focal plane was collimated by the first OAP mirror (MPD00M9-M01, Thorlabs). Then the beam needed to be reduced for measurement. So it was refocused by the second OAP mirror (MPD169-M01, Thorlabs) and re-collimated by a third OAP mirror (MPD129-M01, Thorlabs). At this point, the beam size of the collimated laser was small enough to fit the FROG apparatus' entrance aperture. The FROG traces were retrieved, and the pulse width, pulse front tilt, and the spatial dispersion were calculated with the built-in retrieval algorithm (QuickFrog, Swamp Optics). The focus was parked at the three FOV locations (on-axis, 1.25-mm off-axis, and 2.5-mm off-axis) for the FROG measurement by deflecting the angle of the X-galvo scanner (0, 5, and 10 degrees), corresponding to the angle of 0, 2.5, and 5.0 degrees off-axis at the entrance pupil of the objective. Rotating the X-galvo scanner (instead of the Y-galvo scanner) maximizes the off-axis traveling pathway for the deflected laser beam in the system, and thus provides an upper-limit to any distortions detected.

**Statistics and reproducibility**. For resolution measurements in Fig. 1f, eight or seven beads were measured for each location, and all data points are plotted. As this study is to demonstrate the functionality of a microscopy technique, rather than drawing biological conclusions, replicate experiments were not performed with animals.

**Reporting summary**. Further information on research design is available in the Nature Research Reporting Summary linked to this article.

## Data availability
The data sets reported here are openly available in "figshare" at "https://doi.org/10.6084/m9.figshare.15163914".

## Code availability
The code used in this work is already publicly available as detailed above. If additional materials are required for replication, the authors invite such requests.

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

## Acknowledgements
We thank Ikuko Smith for scientific mentorship and lab leadership. We thank Swamp Optics for the custom prism. We also thank Angelos Loukidis for building the 3D model of the Diesel2p system in Solidworks. This work was supported by the McKnight Foundation, Human Frontier Science Program (RGP0027/2016), the NSF (NeuroNex #1934288 and BRAIN EAGER #1450824) and the NIH (NINDS R01NS091335 and NEI R01EY024294).

## Author contributions
C.H.Y. assembled, aligned, and optimized the system, collected data, analyzed data, and wrote the manuscript. J.N.S. developed the concept, designed the optics, oversaw manufacturing, and collected pilot data. Y.Y. performed mouse surgeries, consulted on imaging, and analyzed data. R.H. optimized the system and collected pilot data. S.L.S. supervised the project.

## Competing interests
The Diesel2p is not patented, and it will not be patented in the future. All designs originating in this report are free for reuse. SLS is a paid consultant for companies that sell multiphoton microscopes. There are no other competing interests.
