## [Peer Review File · Nature Communications]

The previous round of review was completed at another journal

REVIEWER COMMENTS

Reviewer #1 (Remarks to the Author):

The authors have well addressed my previous concerns in term of sub-cellular resolution, FOV and novelty of time multiplexing. I so propose that the manuscript be published in Nature communication.

(I agree on keeping Supplementary Figure 6)

Reviewer #3 (Remarks to the Author):

The authors have adequately addressed my comments and those of the other reviewers in my opinion. I think this paper not only introduces impressive conceptually new technology that may gain wide adoption, but will also serve as a valuable resource for those interested in understanding the applications, advantages, and limitations of various 2p mesoscope systems. It can also serve as a benchmark for further advances.

Sincerely,
Hillel Adesnik

Reviewer #4 (Remarks to the Author):

The authors present a design for a custom two-photon microscope system with a large field of view and dual independent scan engines. The design incorporates numerous advanced optical features, including adaptive optics (AO) for wavefront shaping, temporal multiplexing for simultaneous imaging, and polarization optics for beam recombination, allowing a large imaging volume ($\sim 25 \text{ mm}^2$ FOV) to be addressed at high spatial resolution (lateral FWHM $\sim 1 \mu\text{m}$, axial FWHM $\sim 8 \mu\text{m}$). The design builds on a previous microscope by the same authors (Stirman et al. Nature Biotechnology 2016), with the advantage that the two scan engine arms are independent, allowing imaging parameters to be customized for each imaging region. The instrument is carefully calibrated and nicely presented, and the system design is laudably open-source. This microscope should enable many exciting new experiments for systems neuroscience.

Major comments:

1. It would strengthen the manuscript significantly if the authors would show more examples of correlations measured across spatially highly distinct regions. The authors extol the flexibility of their approach, but the real merit of their design compared with the Thorlabs mesoscope is that you can measure simultaneous activity across spatially widely separated regions. The correlations they do measure [Fig. 2(g)] are from neighboring regions, which mitigates some of the impact of the demonstration. I would have liked to see these correlations across distinct regions.

2. In Figure 1f, it appears that the axial resolution for deeper imaging planes is better than for more superficial planes. Was AO correction adjusted per depth for that figure? Also, the amplitude of the difference in z (from 70 to 500 microns deep) is not achievable by the remote focusing (only ± 60 microns), so the sample appears to have been physically moved. Taking into consideration that the sample is beads in agarose, which is non-scattering (as Reviewer 3 pointed out as well), then this manipulation provides limited insight. The main contributing factor for potential differences in PSF here probably is the distance below the coverslip, and not the amount of scattering tissue the laser is going through. Would one even expect any difference in PSF in such measurements?

3. In the answer to Reviewer 3 point 5, the authors compare the frame rates to the Thorlabs mesoscope. However, they do not discuss the jumping time required if imaging more than 2 ROIs (Figure 3c,d). In a separate place (Supplementary Figure 6) they make a strong case for the benefit of no jumping. Also, they do not take into account the dual-plane module now available for the Thorlabs mesoscope (Tsyboulski, D., Orlova, N., Lecoq, J., & Saggau, P. (2018). MesoScope Upgrade: Dual Plane Remote Focusing Imaging System for Recording of Ca²⁺ Signals in Neural Ensembles. Biophotonics Congress: Biomedical Optics Congress 2018 (Microscopy/Translational/Brain/OTS). doi:10.1364/translational.2018.jw3a.60), which effectively doubles the frame rate (although with limited flexibility of ROI locations)

4. The system is advertised to be designed for two wavelengths, but it is not clear how practical it is to switch between the two - how much realignment/optimization is necessary if using a single tunable laser. This should be added to discussion.

5. Truly simultaneous imaging of two areas would be particularly useful for imaging voltage indicators (only calcium indicators are mentioned in the manuscript). In that case sequential imaging (with delays of 20-30ms between consecutive frames) is just not an appropriate tool, especially if one wants to measure fast-scale correlations between cells in different areas. This could represent a use case offering a real experimental advantage of the current system, and should be mentioned in the discussion.

Minor comments:

1. What is the crosstalk between signals when measuring in overlapping regions, as in Fig 3b? The pulse delay (6.25 ns) isn't too far off the lifetime of GCaMP, so the authors should at least quantify the crosstalk.

2. Supplementary Figure 10 shows the scenario where two separate light sources are used, but the details are not clear - were the two lasers used on separate arms? At what point they were combined and how the pulse/acquisition timing was being managed?

3. When talking about the throughput in terms of pixels/s, it is also important to add the number of pulses they deliver per pixel into the consideration. This number will also be different along the resonant axis.

4. In their answer to Reviewer 3 point 5, they mention the ability to rotate the objective 360 degrees. However, this has certain practical limitations, especially for in vivo experiments, as the rotation is not around the focal point.

5. Figure 2g caption: "(g) Spikes in (f) were used to measure > 17 million cross-correlations the population" - 'the' should be replaced by 'per'?

6. Figure 2 general comment: we do not have enough pixels on the screen and dpi on the paper to properly plot matrices of 6000x6000 neurons.

Dear Editor,

We thank the reviewers for their detailed assessment of our manuscript and their constructive feedback. A detailed description of the resulting changes, and our responses to the comments, are provided below (in black text). We believe that our manuscript is much improved, and now suitable for publication.

Reviewer #1 (Remarks to the Author):

The authors have well addressed my previous concerns in term of sub-cellular resolution, FOV and novelty of time multiplexing. I so propose that the manuscript be published in Nature communication.
(I agree on keeping Supplementary Figure 6)

Thank you for the positive assessment, and your agreement on keeping Supplementary Figure 6.

Reviewer #3 (Remarks to the Author):

The authors have adequately addressed my comments and those of the other reviewers in my opinion. I think this paper not only introduces impressive conceptually new technology that may gain wide adoption, but will also serve as a valuable resource for those interested in understanding the applications, advantages, and limitations of various 2p mesoscope systems. It can also serve as a benchmark for further advances.

Sincerely,
Hillel Adesnik

Thank you for the positive assessment.

Reviewer #4 (Remarks to the Author):

The authors present a design for a custom two-photon microscope system with a large field of view and dual independent scan engines. The design incorporates numerous advanced optical features, including adaptive optics (AO) for wavefront shaping, temporal multiplexing for simultaneous imaging, and polarization optics for beam recombination, allowing a large imaging volume ($\sim 25 \text{ mm}^2$ FOV) to be addressed at high spatial resolution (lateral FWHM $\sim 1 \text{ }\mu\text{m}$, axial FWHM $\sim 8 \text{ }\mu\text{m}$). The design builds on a previous microscope by the same authors (Stirman et al. Nature Biotechnology 2016), with the advantage that the two scan engine arms are independent, allowing imaging parameters to be customized for each imaging region. The instrument is carefully calibrated and nicely presented, and the system design is laudably open-source. This microscope should enable many exciting new experiments for systems neuroscience.

Thank you for the positive assessment.

Major comments:

1. It would strengthen the manuscript significantly if the authors would show more examples of correlations measured across spatially highly distinct regions. The authors extol the flexibility of their approach, but the real merit of their design compared with the Thorlabs mesoscope is that you can measure simultaneous activity across spatially widely separated regions. The correlations they do measure [Fig. 2(g)] are from neighboring regions, which mitigates some of the impact of the demonstration. I would have liked to see these correlations across distinct regions.

We thank the reviewer for this suggestion to calculate activity correlations of neurons in distinct regions that are distant from each other. The Diesel2p system provides both: (1) simultaneous imaging of neurons in distant regions and (2) can also optimize the imaging parameters (e.g., frame rate, region size, and scan pattern) independently in each pathway based on the local neural circuitry and experimental requirements.

The reviewer is correct that this demonstrates an advantage compared to the Thorlabs mesoscope, which uses a single beam that jumps among areas, and this entails a dead time that reduces the temporal resolution of the sampling and decreases the duty cycle for data collection (Supplementary Fig. 6).

The reviewer's comment makes it clear that Fig. 2g was not clearly depicting the distances between neurons from which the correlations were measured. We have added a new panel (Fig. 2h) that provides a clearer depiction. Also, we can elaborate. The correlation matrix in Fig. 2g is computed from the large-field data acquired in Fig. 2c, where the size of the imaging region is 3 mm x 5 mm (across both beams), a field that spans multiple cortical areas. In addition, Fig. 3a further demonstrated the capability to perform independent scanning of two distant regions simultaneously. The largest distance between neurons in this measurement is ~4.4 mm. These data points together argue that the measurement from the neighboring regions was mistakenly conveyed, and demonstrate the capability of the Diesel2p system to access neurons at distant regions simultaneously.

As the reviewer requested, we analyzed the distance-dependence of correlations of neural activity acquired in Fig. 2c, and plotted the results below. When the neural activity was recorded, the mouse was idle in the dark. The figure below shows the average correlation as a function of the distance between neurons ranging from 0 μm to 4000 μm . The correlations are higher at a distance shorter than 40 μm , and quickly decrease beyond 40 μm . As the reviewer also asked for a more proper way (reviewer's comment 6 below) to present the 6000 by 6000 correlation matrix shown in Fig. 2g, we think that this new analysis is a good alternative to present the data properly. Therefore, we have added new analysis as a new panel Fig. 2h.

2. In Figure 1f, it appears that the axial resolution for deeper imaging planes is better than for more superficial planes. Was AO correction adjusted per depth for that figure? Also, the amplitude of the difference in z (from 70 to 500 microns deep) is not achievable by the remote focusing (only ± 60 microns), so the sample appears to have been physically moved. Taking into consideration that the sample is beads in agarose, which is non-scattering (as Reviewer 3 pointed out as well), then this manipulation provides limited insight. The main contributing factor for potential differences in PSF here probably is the distance below the coverslip, and not the amount of scattering tissue the laser is going through. Would one even expect any difference in PSF in such measurements?

We agree with the reviewer that this characterization is not to test and correct the tissue-induced aberration. However, the configuration of the bead measurement preserves the refractive index mismatches from the air-glass-agar (brain) interfaces. The index of refraction change between air and the agar block imparts optical aberrations, especially the spherical aberration, that change with depth of imaging in the agar block. To what extent the aberration depends on the imaging depth is what we examined in this data, similar to prior studies such as Sofroniew et al. 2016 and Stirman et al. 2016. Therefore, AO correction was adjusted at different imaging depths. In our optical model, we optimized the PSF at the depth of 500 microns under the coverslip and with a 150- μm thick coverslip. This optimal PSF at 500- μm depth is not necessarily preserved perfectly at different imaging depths. As the reviewer concluded, our data indeed show the trend that the PSF is the best at 500- μm depth and gets bigger at the shallower depth. Nevertheless, this effect is not significant over the intended range of the two-photon imaging depths (0 ~ 500 μm). Although it is seemingly not a severe issue based on our results, we think that this still worth a characterization, especially for custom-built systems, as we find that it aids in an apples-to-apples comparison across publications in the field.

3. In the answer to Reviewer 3 point 5, the authors compare the frame rates to the

Thorlabs mesoscope. However, they do not discuss the jumping time required if imaging more than 2 ROIs (Figure 3c,d). In a separate place (Supplementary Figure 6) they make a strong case for the benefit of no jumping. Also, they do not take into account the dual-plane module now available for the Thorlabs mesoscope (Tsyboulski, D., Orlova, N., Lecoq, J., & Saggau, P. (2018). MesoScope Upgrade: Dual Plane Remote Focusing Imaging System for Recording of Ca²⁺ Signals in Neural Ensembles. Biophotonics Congress: Biomedical Optics Congress 2018 (Microscopy/Translational/Brain/OTS). doi:10.1364/translational.2018.jw3a.60), which effectively doubles the frame rate (although with limited flexibility of ROI locations)

We now have added text to say that jumping is required if one pathway is imaging more than 2 ROIs in Fig. 3c and 3d. The two pathways are still imaging simultaneously and independently.

As the reviewer suggested in item 5 below, we added in the discussion that no-jumping is advantageous over jumping for dual-region imaging of voltage indicators.

We had cited the paper that the reviewer suggested, and thus had taken into account the dual-plane module of the Thorlabs mesoscope. There are multiple reports of the work, including a preprint and a conference paper. Here is where we had referenced the work: *“Temporal multiplexing of simultaneously scanned beams can increase throughput⁷, and these can have either a fixed configuration⁸⁻¹⁰, or can be reconfigured during the experiment axially¹¹⁻¹³ or axially and laterally^{14,15}.”* Reference #11 is “Tsyboulski, D. et al. Remote focusing system for simultaneous dual-plane mesoscopic multiphoton imaging. bioRxiv, 503052, doi:10.1101/503052 (2018)”.

We like citing the preprint because once it is published in a journal, the preprint will reference that. We acknowledge that the this customized Thorlabs mesoscope can access two regions simultaneously, doubling the frame rate. However, the two regions can only be deployed separately in the axial direction— not laterally— so two cortical areas cannot be imaged simultaneously. In addition, the imaging parameters of these two regions cannot be independently configured. Dual-region imaging positionable in three dimensions and independently configurable are the major distinct features of the Diesel2p system from other mesoscopes.

4. The system is advertised to be designed for two wavelengths, but it is not clear how practical it is to switch between the two - how much realignment/optimization is necessary if using a single tunable laser. This should be added to discussion.

Nothing has to change in the microscope as the Diesel2p system is designed to be achromatic in these two wavelengths. If there are two lasers used simultaneously, the two lasers can be merged with a dichroic mirror before being coupled into the system. In this revision, we have added text to discussion and to the methods to state that *“The optics are achromatic for the 910 nm and 1050 nm wavelength windows, so both*

wavelengths can be used simultaneously in either or both paths without any reconfiguration of the imaging system.”

5. Truly simultaneous imaging of two areas would be particularly useful for imaging voltage indicators (only calcium indicators are mentioned in the manuscript). In that case sequential imaging (with delays of 20-30ms between consecutive frames) is just not an appropriate tool, especially if one wants to measure fast-scale correlations between cells in different areas. This could represent a use case offering a real experimental advantage of the current system, and should be mentioned in the discussion.

Thank you for the suggestion. We now mention that in the discussion, saying: “*The advantage of the Diesel2p’s simultaneous imaging, with zero jumping time, can be critical when imaging very fast dynamics such as neurotransmitter reporters²² and voltage indicators²³ expressed at distant brain areas.*”

Minor comments:

1. What is the crosstalk between signals when measuring in overlapping regions, as in Fig 3b? The pulse delay (6.25 ns) isn't too far off the lifetime of GCaMP, so the authors should at least quantify the crosstalk.

As the reviewer suggested, we characterized the crosstalk between the two multiplexed pathways of the 910-nm wavelength. We performed *in-vivo* imaging of a mouse expressing GCaMP6, and the figure below shows the result. Neurons were (a) imaged with Pathway 1 while blocking excitation in Pathway 2, and (b) vice versa. In the rightmost panels, the image intensity in the blocked pathway is scaled up by 10 times to show the spatial structure of the signal, if there is any. The result shows that the crosstalk is minimal, and there is no structured image bled through into the blocked channel.

This is an interesting point: the independence of the two scan engines desynchronizes any small residual crosstalk to nullify any potential contamination. The lack of structure of the bleed-through is due to the mismatch of the scanning frequencies between the two independent scan engines, as each scan engine has its own resonant scanner whose resonant frequency is slightly different from the other one. Although the blocked pathway may receive bleed-through photons from the unblocked pathway, the image in the blocked pathway is created using a scanner operating at a different frequency so there is little-to-no structure remaining in the image. This result is in contrast to the temporal-multiplexed pathways in a single scan engine system (e.g., our prior Trepan2p system, or the customized dual-plane Thorlabs Mescope work mentioned above), where crosstalk will result in structured images between imaging pathways. We have added this result into the manuscript as Supplementary Figure 11.

2. Supplementary Figure 10 shows the scenario where two separate light sources are used, but the details are not clear - were the two lasers used on separate arms? At what point they were combined and how the pulse/acquisition timing was being managed?

We have now added a diagram in Supplementary Figure 11 to explain (also shown below). A 1040-nm pulsed laser with a 40 MHz repetition rate is introduced into the system and merged with the 910-nm laser using a dichroic mirror. Before being merged with the 910-nm laser beam, the 1040-nm laser was temporally multiplexed by being split into two beamlets, and one beamlet was delayed from the other by ~ 12.5 ns ($1/40\text{MHz}/2$, using two 6.25 ns delay stages). After the beams were merged coaxially with the dichroic mirror, the beams were guided through the same optics downstream in the system, so that each pathway can use the two lasers simultaneously. Two photomultiplier tubes were used to collect the spectrally-separated emission photons excited by the 910-nm and 1040-nm lasers, individually. The signal collected by each photomultiplier tube is demultiplexed based on the laser arrival time, respectively, achieving a dual-band, dual-beamlet, dual-scan engine excitation and dual-channel collection system. In summary, there are four separate ultrafast pulse trains (two wavelengths, each with 2 beamlets) directed across the preparation using two independent scan engines (one for P-polarization beamlets, and one for S-polarization

beamlets). With this configuration, both the 1040-nm laser and the 910-nm laser have simultaneous access to the full FOV of the two independent scan engines using temporally multiplexed foci.

3. When talking about the throughput in terms of pixels/s, it is also important to add the number of pulses they deliver per pixel into the consideration. This number will also be different along the resonant axis.

We now provide the formula to calculate the pulses per pixel as follows. The pulses per pixel is a function of the fill fraction (FF), the number of pixels per line (N), the resonant frequency of the resonant scanner (Freq), and the repetition rate of the laser (Rep), and the position of the pixel on the resonant axis (n). The FF is the ratio between the length of an digitized portion of a scan line and the total length of the line (the far limits of scan lines are typically discarded and/or blanked).

$$\begin{aligned}
 & \text{Pulses per pixel } (n, FF, N, Freq, Rep) \\
 &= \frac{2 * FF}{N} * \frac{Rep}{2 * \pi * Freq * \cos \left\{ \sin^{-1} \left[-FF + \frac{2 * FF}{(N - 1)} (n - 1) \right] \right\}}
 \end{aligned}$$

We plotted the pulses per pixel in the three imaging conditions used in this manuscript below.

We now have added the formula in the method section and the plots as a Supplementary Figure 12.

4. In their answer to Reviewer 3 point 5, they mention the ability to rotate the objective 360 degrees. However, this has certain practical limitations, especially for in vivo experiments, as the rotation is not around the focal point.

We agree with the reviewer that our version of rotation is not as ideal as the rotation around the focal point implemented in other systems, such as the Thorlabs Bergamo II series. We now added this into the discussion: *“In addition, the objective can rotate 360 degrees to work with non-horizontal imaging surfaces. This rotation facilitates imaging in some preparations and could be further improved if the axis of rotation was along the focal plane of the objective.”*

5. Figure 2g caption: “(g) Spikes in (f) were used to measure > 17 million cross-correlations the population” - ‘the’ should be replaced by ‘per’?

Thank you for pointing this out. We corrected it.

6. Figure 2 general comment: we do not have enough pixels on the screen and dpi on the paper to properly plot matrices of 6000x6000 neurons.

We now have added a new plot in Fig. 2 (panel h) to present the data in an alternative form to the correlation matrix in Fig 2g. Fig. 2h (and shown below) shows the average correlation as a function of the distance between neurons, ranging from 0 μm to 4000 μm , thus demonstrating a measurement that is not possible with conventional instrumentation. The correlations are high for neurons within 40 μm of each other, and then they drop to a lower level and more slowly trend towards lower correlations until

about 2500 μm . These measurements demonstrate a unique function of the system, and are readable at conventional display settings.

REVIEWERS' COMMENTS

Reviewer #4 (Remarks to the Author):

The authors have done a good job in dealing with the reviewers' comments, and the manuscript is much improved. I have no further comments.